# Socioeconomic differences in inpatient care expenditure in the last year of life among older people: a retrospective population-based study in Stockholm County

Megan Doheny [1,2] Pär Schön,[2] Nicola Orsini [1] Anders Walander,[3] Bo Burström,[1] J Agerholm[2]

[1] Global Public Health, Karolinska Institutet, Stockholm, Sweden
[2] Aging Research Center, Department of Neurobiology, Care Sciences and Society, Karolinska Institutet and Stockholm University, Stockholm, Sweden
[3] Center for Epidemiology and Community Medicine, Region Stockholm, Stockholm, Sweden

**Correspondence to**
Dr Megan Doheny;
megan.doheny@ki.se

## ABSTRACT

**Objectives** To investigate the association between inpatient care expenditure (ICE) and income group and the effect of demographic factors, health status, healthcare and social care utilisation on ICE in the last year of life.

**Design** Retrospective population-based study.

**Setting** Stockholm County.

**Participants** Decedents ≥65 years in 2015 (N=13 538).

**Outcome** ICE was calculated individually for the month of, and 12 months preceding death using healthcare register data from 2014 and 2015. ICE included the costs of admission and treatment in inpatient care adjusted for the price level in 2018.

**Results** There were difference between income groups and ICE incurred at the 75th percentile, while a social gradient was found at the 95th percentile where the highest income group incurred higher ICE (SEK45 307, 95% CI SEK12 055 to SEK79 559) compared with the lowest income groups. Incurring higher ICE at the 95th percentile was driven by greater morbidity (SEK20 333, 95% CI SEK12 673 to SEK29 993) and emergency department care visits (SEK77 995, 95% CI SEK64 442 to SEK79 549), while lower ICE across the distribution was associated with older age and residing in institutional care.

**Conclusion** Gaining insight into patterns of healthcare expenditure in the last year of life has important implications for policy, particularly as socioeconomic differences were visible in ICE at a time of greater care need for all. Future policies should focus on engaging in advanced care planning and strengthening the coordination of care for older people.

## STRENGTHS AND LIMITATIONS OF THIS STUDY

⇒ The quantile regression has flexible assumptions compatible with cost data and assesses the relationship between factors across the distribution of inpatient care expenditure (ICE).

⇒ In this study, it was only feasible to measure ICE; however, this does not reflect the total care costs that are accrued during the last year of life.

⇒ Measuring need of care is difficult, particularly, during the last year of life as needs change.

proportion of an individual's lifetime healthcare expenditure occurs during the last year of life.[5]

Patterns of healthcare utilisation in the last year of life often involve high rates of acute hospital-based services.[6] Congruently, a country comparison study of the composition of end-of-life (EOL) expenditure across nine different countries showed that expenditure on hospital-based inpatient care accounted for the largest proportion of total expenditure, followed by the utilisation of social care.[7] Additionally, a Swedish study observed that inpatient care specialties accounted for 80% of state expenditure on healthcare in the last year of life.[8]

Often high spending on healthcare among older persons is not due to the use of more expensive life-saving treatments, but rather due to spending on treating persons with multiple chronic conditions.[5] Generally, healthcare utilisation and as such healthcare expenditure is largely influenced by the need of care, and health status which varies by socioeconomic position (SEP). Frequently, older persons with lower SEP experience poorer health and have greater need of hospital-based care at the EOL which would

## INTRODUCTION

An ageing population is often considered the main driver of increasing healthcare expenditure due to increasing prevalence of multimorbidity (2+chronic conditions) and more complex health problems.[1] However, age itself is not the main driver of healthcare expenditure but rather proximity to death, as a large proportion of all deaths occur among those 65 years and older,[2–4] and a substantial



lead to decedents with lower SEP incurring higher inpatient care expenditure (ICE).[9] However, previous studies have observed socio-economic differences in healthcare expenditure, where persons with higher SEP have higher expenditure compared with those with lower SEP.[8 10–13]

The Swedish healthcare and social care system is universal and primarily financed through taxes collected by the regions and municipalities. Additionally, the provision of care is based on the principle of 'equal access for equal need' regardless of an individual's age, sex or economic resources. However, the organisation of care is decentralised where the 21 regions are responsible for health and medical care and the 290 municipalities are responsible for social care for older people (home help services and institutional care).[14] Patient fees account for a small fraction of healthcare costs (3%–5%), and moreover, there are cost ceilings for out-of-pocket patient fees, set at SEK1105 on healthcare visits during a 12-month period, when this threshold is exceeded, there is no further patient fees for the proceeding 12 months. The out-of-pocket costs of an inpatient care stay is SEK100 per day per adult.[14] So accordingly, a recent report from the Organisation for Economic Co-operation and Development (OECD) showed that Sweden has one of the highest average healthcare expenditures per capita, €4676. On the other hand, there is on average 2.1 hospital beds per 1000 persons in Sweden which is among the lowest in OECD.[15] In municipal institutional care there is onsite health and medical care being provided by nurses, assistant nurses and primary healthcare (PHC) doctors, in addition to care personnel that provide daily personal care (bathing, eating and hygiene), social activities and companionship in institutional care settings.[16] Home help services can be offered around the clock and provides two main types of support: household tasks (eg, cleaning and laundry, shopping, cooking or meals on wheels) and personal care. In parallel to the reduction in hospital beds, there has been a similar decrease in the number of places in municipal institutional care, in Sweden. These changes have been brought about as cost containment strategies driven by the 'Ageing in place' policy, which has resulted in an increasing number of older persons dependent on receiving care and support in their own homes.[16]

Increasing the knowledge of the drivers of ICE and investigating whether there are socioeconomic differences is important considering the ageing population with greater care needs and the goal of providing equitable care. This study aimed to investigate the association between ICE and SEPand the effect of other sociodemographic factors, health status, healthcare and social care utilisation on ICE in the last year of life.

## METHODS

The study population was retrospectively identified, including those 65 years and older that died in the calendar year 2015 in Stockholm County (N=13 538). The last year of life was defined as the month of death plus the 12 months preceding the month of death. The Cause of Death register used to identify the study population using the year and month, to measure the age of death, place of death and the underlying cause of death.[17] The place of death is categorised into dying in a hospital, institutions/specialised geriatric clinics, private residence or other. The underlying cause of death categories were cancer-related, cardiovascular-related, neurodegenerative-related (including dementia, Alzheimer's, Parkinson's disease) and other causes.[18]

### Data sources

The Region Stockholm Healthcare Administrative Databases (VAL by Swedish acronym) inpatient care register was used to measure the outcome ICE, which included the costs of admission (planned and unplanned) and treatment in inpatient care for acute somatic, geriatric, surgical and psychiatric departments. ICE was calculated for 2014 and 2015, adjusted for the price level in 2018 in SEK, (SEK10=€0.98). The outcome was total ICE for each decedent measured as their total inpatient expenditures accrued from their month of death plus 12 months preceding death.

Sociodemographic characteristics of decedents were obtained from the Longitudinal Integration database for Health Insurance and Labour Market Studies (LISA). This register contains a collection of variables from different population registers linked individually via encrypted serial numbers. We measured sex, country of birth, living situation and income in LISA. Sex was grouped as male and female. Country of birth was dichotomised as born in Sweden or born outside of Sweden. Living situation was measured for decedents living in the community and categorised as cohabiting or living alone. SEP was measured using income and assessed using the net annual equalised individual household income from 2012 and then ranked into income quintiles. Those missing a measure for income were excluded from the main analysis (n=56).

Healthcare utilisation during the last year life was obtained from VAL. PHC use was measured by the number of visits to general practitioner (GPs) in outpatient care, categorised into 0–5 visits, 6–10 visits and >10 visits. Emergency department (ED) visits were defined as a registered emergency care visit to acute care hospitals in Stockholm County, and categorised into 0–1 visits, 2–3 visits and 4+ visits (frequent ED use). Both measures included in the regression analyses as continuous count variables.

Home healthcare utilisation was measured in VAL as a period of being enrolled in receiving a period of basic or advanced home healthcare provided in a patient's home. There are two levels, basic and advanced which are provided based on medical need. Basic home healthcare is provided to those that require simpler medical interventions up to nursing level, such as assistance with taking medication, dressing of wounds, catheter replacement or to those who have difficulties visiting GPs. Basic home healthcare is free for all patients enrolled and those 85 years and older. Advanced home healthcare is provided

to seriously ill patients in need and involves medical procedures performed at home rather than in a hospital setting, often provided to cancer patients or those with complex EOL care needs and is not subject to patient fees.[19]

To measure morbidity prior to death we calculated the Charlson Comorbidity Index (CCI) using registered diagnoses in VAL prior to the last year of life. The CCI assigns scores ranging from 1 to 6 to morbidities based on the severity of illness and risk of death,[20] the CCI was calculated per decedent and was described in categories: 1–2 score, 3–4 score and 5+ score, and was included as a continuous variable in regression analysis.

The utilisation of municipal social care was measured in the Swedish Social Services Register which collects data on use of social care on monthly basis. We identified those receiving home help (personal care and/or domestic services) in their own homes as well as individuals registered as living in an institution for the entire 12-month period. Additionally, there was a group of individuals that were receiving home help services in their own homes in the community, who moved into institutional care during their last year of life, hereafter, this group is referred to as the transition group. In the regression analysis the utilisation of home help and institutional care was measured by number of months of use in the last year of life.

### Patient and public involvement
The data used in this study were based on encrypted personal numbers so that the individuals in the study population are not identifiable.

### Statistical analysis
The quantile regression (QR) was selected to investigate how ICE varied between income groups (SEP) and to identify factors that affect the ICE incurred in the last year of life, because the dependent variable ICE has a positively skewed distribution with extreme outliers.[21 22] The QR model estimates the change in a specified quantile (ie, percentiles) of the conditional distribution of ICE due to a unit change in the independent variable. The QR was used to assess which independent variables are associated with quantiles of expenditure and has been recommended for the analysis of expenditure outcomes compared with alternative approaches in previous studies.[23–25] We estimated ICE at the 50th, 75th and 95th percentiles, adjusting for the following explanatory variables (income, sex, age, country of birth, CCI score, ED use, home healthcare and municipal social care use). These percentiles were selected as it was presumed that the rate of change in ICE per unit change in an explanatory variable ($\Delta ICE/\Delta x$) would be progressively greater in the higher percentiles. There was n=2912 decedents who incurred zero ICE in the last year of life. A logistic regression model was used assess whether there were socio-economic differences among those that incurred zero ICE in the last year of life.

## RESULTS
There were N=13 538 decedents included in the study population described in table 1. Most decedents (51.7%) were 85+ years, (54.2%) female and (25.7%) in income group 3. Most decedents in the community were living alone (63.4%), and 18.5% were born outside of Sweden. There were 38.8% of decedents who had a cardiovascular-related death and 25.3% with a cancer-related underlying cause of death. During the last year of life, decedents had on average 4.5 PHC visits and 2.7 ED visits. Further, 28.6% of decedents frequently attended ED care (4+visits) in the last year of life. Most decedents used municipal social care, 32.5% received home help services, 28.3% were in institutional care and 10.5% had transition from receiving home help into an institution during the last year of life.

Those in the lowest income group incurred higher costs at the 50th percentile (SEK139 401) compared with other income groups, while at the 75th and 95th percentile those in the highest income group incurred higher ICE (SEK287 722 at 75th) and (SEK669 124 at 95th) compared with the lowest income group (SEK281 968 SEK at 75th) and (SEK633 349 SEK at 95th), respectively. There was an inverse relationship between age and ICE, where those 85+ years incurred lower ICE compared with those 65–74 years. Females incurred lower ICE compared with males at the 50th, 75th and 95th percentiles, a similar pattern could be observed among those living alone and those born outside of Sweden.

Decedents that had more visits to PHC and other outpatient care services incurred higher ICE. Those receiving basic home healthcare incurred higher ICE compared with those receiving advanced home healthcare. Decedents receiving home help services or that transitioned to institutional care incurred higher ICE compared with those independent. Persons with cancer-related deaths incurred higher ICE at the 50th, 75th and 95th percentiles, while those with neurodegenerative-related cause of death incurred lower ICE. Most decedents (43.2%) died in hospital, 25.5% died in institutional care and 14.2% died in a private residence. Those that died in hospital incurred higher ICE compared with the other places of death.

### Quantile regression
Table 2 contains the estimates from the QR analysis. We observed socioeconomic differences in ICE incurred at the 75th and 95th percentile by income group. Those in the second, third-income and fourth-income group incurred higher ICE at the 75th percentile compared with the lowest income group. There was a social gradient in the ICE incurred at the 95th percentile as those in higher income groups incurred higher ICE. Those in the highest income group incurred (SEK45 307, 95% CI SEK12 055 to SEK79 559) higher ICE compared with those in the lowest income group at the 95th percentile.

Increasing age in years was associated with lower ICE at the 50th, 75th and 95th percentile. There were no significant differences between sexes. Decedents born outside

**Table 1** Description of decedents 65+ years, and the distribution of of inpatient care expenditure (ICE) in the last year of life

| | | Percentiles of ICE (SEKs) | | |
| --- | --- | --- | --- | --- |
| | N (%) | (Median) 50th percentile | 75th percentile | (Highest) 95th percentile |
| | 13 538 (100) | 129 886 | 279 152 | 644 596 |
| Income group | | | | |
| Group 1 (lowest) | 2647 (19.6) | 139 401 | 281 968 | 633 349 |
| Group 2 | 3160 (23.3) | 131 404 | 284 388 | 656 418 |
| Group 3 | 3475 (25.7) | 120 833 | 260 599 | 617 288 |
| Group 4 | 2517 (18.6) | 132 311 | 291 504 | 660 780 |
| Group 5 (highest) | 1683 (12.4) | 132 203 | 287 722 | 669 124 |
| Age in years | 83.8 ± 9.0 | | | |
| 65–74 years | 2672 (19.7) | 161 587 | 358 466 | 903 567 |
| 75–84 years | 3875 (28.6) | 158 874 | 321 478 | 721 371 |
| 85+ years | 6991 (51.7) | 109 658 | 231 026 | 506 885 |
| Sex | | | | |
| Male | 6204 (45.8) | 147 442 | 306 065 | 702 300 |
| Female | 7334 (54.2) | 116 217 | 257 460 | 591 401 |
| Country of birth | | | | |
| Sweden | 10 963 (81.5) | 131 695 | 281 814 | 647 807 |
| Outside of Sweden | 2489 (18.5) | 128 031 | 275 844 | 624 877 |
| Living situation | | | | |
| Cohabiting | 3544 (36.6) | 168 215 | 322 015 | 700 553 |
| Alone | 6132 (63.4) | 115 251 | 259 849 | 613 316 |
| Healthcare utilisation | | | | |
| Average no PHC visits | 4.5 ± 6.9 | | | |
| 0–5 PHC visits | 9753 (72.0) | 104 608 | 239 889 | 585 186 |
| 6–10 PHC visits | 2011 (14.9) | 191 834 | 345 298 | 735 614 |
| >10 PHC visits | 1774 (13.1) | 216 465 | 387 845 | 804 472 |
| Average no ED visits | 2.7 ± 3.0 | | | |
| 0–1 ED visits | 5460 (40.3) | 25 433 | 88 392 | 299 568 |
| 2–3 ED visits | 4208 (31.1) | 156 467 | 248 957 | 522 653 |
| >4 ED visits | 3870 (28.6) | 317 664 | 481 791 | 870 365 |
| Home health care | | | | |
| Basic care | 3602 (26.6) | 225 700 | 382 229 | 759 464 |
| Advanced care | 1651 (12.2) | 202 885 | 355 234 | 718 165 |
| Social care utilisation | | | | |
| Independent | 3863 (28.5) | 125 057 | 269 586 | 696 304 |
| Home help | 4393 (32.5) | 203 872 | 359 895 | 721 371 |
| Institutional care | 3862 (28.5) | 41 823 | 130 698 | 392 649 |
| Transition | 1420 (10.5) | 202 336 | 346 695 | 710 872 |
| Months with home help | 8,2 ± 4.9 | | | |
| Months in institutional care | 9.0 ± 3.8 | | | |
| Health status and underlying cause of death | | | | |
| CCI score | 3.1 ± 2.6 | | | |
| 0–2 score | 6878 (50.8) | 60 216 | 163 026 | 451 067 |
| 3–4 access | 2873 (21.2) | 188 170 | 341 629 | 697 367 |

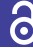

Table 1 Continued

| | N (%) | Percentiles of ICE (SEKs) | | |
| | | (Median) 50th percentile | 75th percentile | (Highest) 95th percentile |
| --- | --- | --- | --- | --- |
| 5+score | 3787 (28.0) | 236 137 | 406 819 | 805 941 |
| Cause of death | | | | |
| Cancer related | 3425 (25.3) | 195 908 | 343 042 | 681 738 |
| Cardiovasuclar-related | 5221 (38.6) | 119 816 | 274 621 | 654 265 |
| Neurodegenerative related | 1995 (14.7) | 45 393 | 143 758 | 396 939 |
| Other | 2897 (21.4) | 132 820 | 288 010 | 686 942 |
| Place of death | | | | |
| Hospital | 5853 (43.2) | 203 204 | 362 355 | 780 051 |
| Institution/specialised geriatric clinic | 4807 (25.5) | 69 464 | 192 532 | 484 093 |
| Private residence | 1918 (14.2) | 68 809 | 209 693 | 552 571 |
| Other | 960 (7.1) | 85,155 | 240 580 | 600 078 |

Variables living situation and home healthcare utilisation are measures applicable to those living in the community for in the last 12 months of life (n=9676).
CCI, Charlson Comorbidity Index; ED, emergency department; PHC, primary healthcare; SEK, Swedish Kronor.

of Sweden incurred lower ICE at the 50th (SEK–2736, 95% CI SEK–4367 to SEK–1104) and at the 75th percentile (SEK–11 318, 95% CI SEK–15 785 to SEK–6851) compared with decedents born in Sweden. Greater morbidity and more visits to ED care were positively associated with incurring higher ICE. ED visits were a significant driver for higher ICE, an increase in the number ED visits was associated with (SEK71 995, 95% CI SEK64 442 to SEK79 549) higher ICE at the 95th percentile. Those receiving basic home healthcare incurred higher ICE at the 50th and 75th percentiles, there was similar pattern with advanced home healthcare but slightly lower at the 75th percentile (SEK34 179, 95% CI SEK21 050 to SEK51 029). Months in institutional care were associated with incurring lower ICE across the distribution, while additional months with home help were associated with lower ICE (SEK–4216, 95% CI SEK–71 386 to SEK–1045) at the 95th percentile

### Zero expenditure
There were 2912 decedents that incurred zero ICE, 56.6% of decedents with zero costs resided in institutional care and 29.9% lived independently. Those with zero cost living in the community had an average age of death of 78.4 years, were mostly male, 26.5% born outside of Sweden, used more PHC than ED care in the last year of life. Most of those with zero ICE (53.2%) had cardiovascular-related death and 53.6% died in private residence. Table 3 shows models estimating odds of incurring zero ICE among decedents that were living in the community, there were no socioeconomic differences observed in the odds of zero ICE, while those born outside of Sweden and had a cardiovascular-related underlying cause of death had higher odds of zero ICE. Older age and the utilisation of

healthcare and social care were associated lower odds of having zero ICE.

### DISCUSSION
This study investigated the socioeconomic differences in ICE in the last year of life and assessed the effect of other demographic factors, health status, healthcare and social care utilisation had on ICE. We observed that decedents in the higher income group incurred higher ICE at the 75th percentile, and there was a social gradient in ICE at the 95th percentile with higher ICE in the higher income groups. Older age was associated with incurring lower ICE across the distribution. Greater morbidity and visits to ED care were positively associated with incurring higher ICE across the distribution. Months in institutional care was associated with incurring lower ICE overall, while months with home elp were associated with lower ICE at the 95th percentile.

Based on findings from earlier studies, we expected that older people with lower SEP due to greater need would use more healthcare and incur greater ICE.[8–13 21] Previous studies from the UK observed that persons in the most deprived quintile incurred higher inpatient care costs than the least deprived,[12] similar findings were observed in other studies on those 65 years and older.[11 13] Our findings are akin with a previous Swedish study that found the persons in the highest-income group had higher healthcare expenditure in the last year of life compared with the lowest.[8] In contrast to this study, Hanratty et al[8] included adults of all ages and measured total healthcare expenditure (inpatient and outpatient costs) and yet, we



**Table 2** Quantile regression (QR) estimates conditional on the 50th (median), 75th and 95th (high-cost patients)

| | 50th percentile | | 75th percentile | | 95th percentile | |
|---|---|---|---|---|---|---|
| | Coefficient (SEK) | 95% CI | Coefficient (SEK) | 95% CI | Coefficient (SEK) | 95% CI |
| Income group | | | | | | |
| Group 1 (lowest) | Ref | | Ref | | Ref | |
| Group 2 | 1925 | −474 to 4323 | 10032 | 4714 to 15350 | 41720 | 14824 to 68615 |
| Group 3 | −520 | −3201 to 2161 | 9354 | 3273 to 15436 | 43252 | 19263 to 67240 |
| Group 4 | 304 | −2468 to 3076 | 11704 | 4495 to 18913 | 42762 | 11875 to 73650 |
| Group 5 (highest) | −912 | −3805 to 1981 | 8360 | −817 to 17537 | 45307 | 12055 to 78559 |
| Age in years | −304 | −469 to 139 | −1672 | −2154 to −1190 | −9761 | −12070 to −7452 |
| Sex | | | | | | |
| Male | Ref | | Ref | | Ref | |
| Female | 2635 | −145 to 5414 | 5126 | −1384 to 11636 | 11614 | −11220 to 34449 |
| Country of birth | | | | | | |
| Sweden | Ref | | Ref | | Ref | |
| Outside of Sweden | −2736 | −4367 to −1104 | −11318 | −15785 to −6851 | −21054 | −41726 to −382 |
| CCI score | 12358 | 11180 to 13537 | 16720 | 14999 to 18441 | 20333 | 12673 to 27993 |
| ED visits | 42586 | 41002 to 44170 | 57362 | 54463 to 60260 | 71995 | 64442 to 79549 |
| Home-healthacre | | | | | | |
| None | Ref | | Ref | | Ref | |
| Basic care | 31017 | 23791 to 38243 | 36040 | 21050 to 51029 | 15299 | −23046 to 55643 |
| Advanced care | 11412 | 2777 to 20048 | 34179 | 20382 to 47976 | −32,009 | −82581 to 18563 |
| Months of home-help | 475 | −54 to 1005 | 456 | −677 to 1589 | −4216 | −7386 to −1045 |
| Months of institutional care | −945 | −1117 to −774 | −2074 | −2741 to −1408 | −11593 | −15136 to −8050 |
| Intercept | 24620 | 11015 to 38224 | 163558 | 122071 to 205545 | 1061896 | 855819 to 1267972 |

Of the distribution of inpatient care expenditure in the last year of life among all decedents 65+ years in Stockholm county.
QR model adjusted for income group (ref=lowest income group 1), age in years, sex (ref=male), country of birth (ref=Sweden), CCI score, ED visits, Home healthcare (ref=no care in the community, months in institutional care and months receiving home help.
CCI, Charlson Comorbidity Index; ED, emergency department; SEK, Swedish Kronor.

observed similar socioeconomic differences despite these deviations in study population and outcome.

Poorer health and functioning in the last year of life should be the main determinant of hospital use and subsequent expenditure on care. As a systematic review and meta-analysis found that socioeconomic differences in EOL healthcare expenditure can vary based on adjustment for need of care.[26] We took this into consideration and included the CCI score as an indicator of 'need' measured prior to the last year of life. The CCI score has been demonstrated to be effective at predicting persons who will incur high healthcare costs.[27] The socioeconomic differences were observed at the top 5% of the distribution and remained after adjusting for need, but whether these differences are indicating inequities in access or treatment cannot be determined in this study.

The socioeconomic differences observed in this study may have many explanations. It could be due to more affluent and better educated individuals, or their families being better equipped to navigate the healthcare system and advocate for more extensive or expensive EOL care. However, this prorich bias of hospital care has had mixed findings, a Scottish study that found there were no differences in the costs incurred by SEP once hospitalised, though differences were observed in when persons from more deprived areas reached hospital care.[28] Further, a Swedish study found that decedents 65+ years with tertiary education were more likely to die in hospital compared with those primary education.[29] This raises the questions about the appropriateness of EOL care in terms of the socioeconomic differences observed in ICE but also that most decedents died in hospital which is not the preferred place of death, as a systematic review reported that people prefer to die in their own homes, even as illness progresses.[30] Although, most community-dwelling decedents who incurred zero ICE in their last year of life died in private residence, it is difficult to discern whether this was their preference and if they received appropriate care in the last year of life.

Age was associated with incurring lower ICE in the last year of life, this finding is consistent with previous studies[3 4 6–13] and in line with the 'Red Herring' theory, which stipulates that as age increases healthcare expenditure decreases and social care expenditure increases.[2 3] Decedents residing in institutional care incurred lower ICE and a large proportion incurred zero ICE. This result

**Table 3** Logistic regression estimates the odds of incurring zero inpatient care costs in the last year of life

| | All zeroes* | Community-dwelling decedents† | | | | | |
| --- | --- | --- | --- | --- | --- | --- | --- |
| | | | Model 1‡ | | | Model 2§ | |
| | N=2912 | N=1254 | OR | 95% CI | | OR | 95% CI |
| **Income group** | | | | | | | |
| (Lowest) group 1 | 17.2% | 20.9% | ref | | | ref | |
| Group 2 | 24.1% | 21.5% | 1.02 | 0.77 | 1.36 | 0.96 | 0.74 | 1.25 |
| Group 3 | 27.9% | 24.6% | 0.93 | 0.70 | 1.24 | 0.90 | 0.69 | 1.18 |
| Group 4 | 18.5% | 19.4% | 0.95 | 0.70 | 1.28 | 0.95 | 0.72 | 1.26 |
| (Highest) group 5 | 12.2% | 13.6% | 1.01 | 0.72 | 1.40 | 0.99 | 0.73 | 1.35 |
| **Age in years** | 84.9 (9.9) | 78.6 (9.1) | | | | | |
| 65–74 years | 19.4% | 39.6% | ref | | | ref | |
| 75–84 years | 22.7% | 30.3% | 0.86 | 0.68 | 1.07 | 0.87 | 0.71 | 1.08 |
| 85+ years | 57.8% | 30.1% | 0.71 | 0.56 | 0.90 | 0.80 | 0.64 | 1.00 |
| **Sex** | | | | | | | |
| Male | 39.2% | 54.3% | ref | | | ref | |
| female | 60.8% | 45.7% | 0.90 | 0.75 | 1.09 | 0.96 | 0.80 | 1.15 |
| **Living situation** | | | | | | | |
| Cohabiting | | | ref | | | ref | |
| Living alone | 69.1% | 69.1% | 1.13 | 0.91 | 1.41 | 1.23 | 1.00 | 1.50 |
| **Country of birth** | | | | | | | |
| Sweden | | | ref | | | ref | |
| Outside of Sweden | 19.5% | 26.5% | 1.42 | 1.13 | 1.77 | 1.39 | 1.13 | 1.70 |
| **CCI score** | 1.1 (1.6) | 1.0 (1.9) | 0.59 | 0.56 | 0.62 | | |
| **Healthcare utilisation** | | | | | | | |
| Average no ED visits | 0.3 (0.7) | 0.4 (0.8) | 0.17 | 0.15 | 0.19 | 0.14 | 0.13 | 0.16 |
| Average no PHC visits | 1.9 (4.8) | 4.3 (6.5) | 0.99 | 0.97 | 1.00 | | |
| **Home healthcare** | | | | | | | |
| None | | | ref | | | ref | |
| Basic | 5.2% | 12.0% | 0.55 | 0.42 | 0.71 | 0.44 | 0.35 | 0.57 |
| Advanced | 4.3% | 10.0% | 1.04 | 0.77 | 1.42 | 0.70 | 0.51 | 0.95 |
| **Underlying cause of death** | | | | | | | |
| Other | 21.6% | 25.2% | | | | ref | |
| Cancer related | 8.4% | 15.3% | | | | 0.29 | 0.22 | 0.39 |
| Cardiovascular related | 41.3% | 53.2% | | | | 1.30 | 1.05 | 1.60 |
| Neurodegenerative related | 28.7% | 6.3% | | | | 1.21 | 0.84 | 1.72 |
| **Place of death** | | | | | | | |
| Hospital | 7.9% | 16.3% | | | | | |
| Institution/geriatric specialised clinic | 57.1% | 10.8% | | | | | |
| Private residence | 23.5% | 53.6% | | | | | |
| Unknown | 11.5% | 19.3% | | | | | |
| **Social care utilisation** | | | | | | | |
| Independent | 29.9% | 68.8% | | | | | |
| Home help | 10.5% | 24.2% | | | | | |
| Institutional care | 56.6% | | | | | | |
| Transition | 3.0% | 7.0% | | | | | |

**Table 3** Continued

| | All zeroes* | Community-dwelling decedents† | | | | | |
|---|---|---|---|---|---|---|---|
| | | | Model 1‡ | | | Model 2§ | | |
| | N=2912 | N=1254 | OR | 95% CI | | OR | 95% CI | |
| Months with home help | | | 0.98 | 0.96 | 1.00 | 0.97 | 0.95 | 0.99 |

Logistic regression models stratified into the group of persons with zero costs living in the community n=1254 for the most of their last year of life.
*Column 1 the description of those with zero ICE in the last year of life in proportions (%) and mean and SD.
†Community-dwelling decedents refers all individuals living in their own in the community with or without home help services.
‡Model 1 is adjusted for (income group, age groups, sex, country of birth, living situation, phc visits, ED visits, home healthcare, CCI score, months with home help).
§Model 2 is adjusted for (income group, age groups, sex, country of birth, living situation, PHC visits, ED visits, home-healthcare, underlying cause of death, months with home help).
CCI, Charlson Comorbidity Index; ED, emergency department; ICE, inpatient care expenditure; PHC, primary healthcare; ref, reference group.

might be due to the around-the-clock presence of health professionals and caregivers in institutional care settings, which may facilitate better EOL care. Previous studies observed that independent community-dwelling decedents were more frequently hospitalised and had higher healthcare costs compared with those in institutional care after standardising for similar care needs.[31] Similarly, a Swedish study found that older people living in their own homes were hospitalised more frequently in the last 10 weeks of life.[32]

Older people are now increasingly 'ageing in place', and their care needs shall be met at home. There is a greater need for advanced care planning near the EOL, as most decedents in were receiving municipal social care and even those receiving home healthcare incurred high ICE. A Swedish report comparing healthcare systems in 10 different countries, highlighted that Sweden had deficits in care planning at the EOL.[33] ED visits were a driver of higher ICE, possibly due to subsequent unplanned hospitalisations and treatment in inpatient care.[32] However, if care received in the home is sufficiently meeting needs, as found in a systematic review, receiving appropriate palliative home-based care should lower ED use among dying patients.[34] Nevertheless, a previous Swedish study found that receiving home-healthcare was associated with frequent ED use.[35]

Additionally, care transitions that occur in the last year of life are stressful for patients and family members as well as being costly and difficult to organise in the care system. A Canadian study found that high-cost acute care users had multiple care transitions during the last year of life and had longer hospital stays because of lack of places in institutional care or due to lack of availability of homecare to discharge patients.[36] The provision of social care is essential in the last year life and can reduce inpatient care costs. The deinstitutionalisation trend might leave some patients in precarious living situations during a difficult time, as the overall length of stay in institutions is decreasing with a large proportion of people moving into institution and dying shortly afterward, in Sweden.[37]

### Strengths and limitations

A strength of this study is the use of high-quality register data that allowed us to retrospectively follow our study population and measure their ICE and their utilisation of other health and social care services in last year of life. The QR model allowed us to assess the relationship between SEP and other factors across the distribution of ICE from lowest to the highest, as the QR model has flexible assumptions.[21 22] The Swedish Cause of Death Register provides complete coverage of the population, and we have measures for all decedents.[18] However, as older persons often experience multi-morbidity deciphering the exact cause of death can be difficult especially among the very old.[38] Therefore, we also included the CCI score to indicate need,[20] though we are limited in our ability to fully measure an individual's need and the extent that these needs change during the last year of life.[5 27] Another limitation of this study is that we only focused on expenditure on inpatient care, which does not provide a complete overview of total costs accrued in the EOL, as social care accounts for a substantial proportion of EOL care and there are other costs such as for outpatient visits and pharmaceutical drugs.[4 7] Future research should use a longitudinal approach and explore the socioeconomic differences in patterns of health and social care services in the last year of life and the years prior to understand how socioeconomic differences can emerge.

### CONCLUSIONS

Most deaths occur in older age, and now most older people with complex health problems are receiving care in their own homes. Gaining insight into patterns of healthcare spending in the last year of life has important implications for policy, as socioeconomic differences were visible

in ICE at a time of greater care need for all, indicating inequity in EOL care. Future policies should focus on engaging in advanced care planning and strengthening the coordination of care in older people homes.

**Contributors** MD, PS, and BB conceived the idea for, participated in the design and the coordination of this study. AW, JA and NO provided advice on data collection, variable measures and statistical analysis. MD created the dataset used in the study, performed the statistical analysis and drafted the manuscript. JA, NO, PS and BB were all involved in the interpretation of results and commented on manuscript drafts. All authors read and approved the submitted manuscript. JA is the gaurantor of this study.

**Funding** The authors disclosed receipt of the following financial support for the research, authorship and/or publication of this article: The research leading to these results was carried out as part of the Social Inequalities in Ageing (SIA) project, funded by NordForsk project no. 74637, and funding was also provided by the Swedish Research Council for Health, Working Life and Welfare (Forte), grant numbers 2017-02155 and 2017-04431.

**Competing interests** None declared.

**Patient and public involvement** Patients and/or the public were not involved in the design, or conduct, or reporting, or dissemination plans of this research.

**Patient consent for publication** Not applicable.

**Ethics approval** Ethical permissions were obtained from the Regional Ethics Review Board of Stockholm (Dnr 2016/299-31).

**Provenance and peer review** Not commissioned; externally peer reviewed.

**Data availability statement** No data are available. The data used will not be shared, as the materials are based on individual data which cannot be shared in accordance with Swedish law and GDPR. The data used in this article will be made available wherever legally and ethically possible.

**ORCID iDs**
Megan Doheny http://orcid.org/0000-0002-5640-1239
Nicola Orsini http://orcid.org/0000-0002-2210-5634

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
