## [Reviewer comments · BMJ Open]

ARTICLE DETAILS

TITLE (PROVISIONAL)	Socio-economic differences in inpatient care expenditure in the last year of life among older people: A retrospective population-based study in Stockholm County
AUTHORS	Doheny, Megan; Schön, Pär; Orsini, Nicola; Walander, Anders; Burström, Bo; Agerholm, J

VERSION 1 – REVIEW

REVIEWER	Nanako Tamiya University of Tsukuba, Health Services Research
REVIEW RETURNED	28-Feb-2022

GENERAL COMMENTS	I read with high interest. This paper is well-written and informative for aging society. I have some comments to be revised. p4 line46 To understand the background behind this paper that high income is related to high ICE, more information will be needed on how much is " a small fraction" is there any difference by income level? p5 etc Is there any information on the place of death and how long they stay at the hospital? Is the hospital stay included in Months of institutional care? They are important information. p6 line 5 "Country of birth " should be like " birth outside or inside Sweden" because no info about other countries p7 line54 What are others in social care? p9 line19 the result that advanced home healthcare users have lower ICE is very interesting, In the advanced home healthcare, more medical care may be offered and can be avoided hospitalization? More explanation will be helpful including the payment of these two types of home care. line 21 " home-help users -- compared to those independent" This comparison is right? Home-help users should be always independent? p15 line 38 If there is any previous study about income level and ACP, it may be helpful.
--

REVIEWER	S Antoniu Grigore T. Popa University of Medicine and Pharmacy
REVIEW RETURNED	04-Apr-2022

GENERAL COMMENTS	this is a very interesting analysis. i would suggest to add some more results 1. analysis for place of death hospital versus home/home care institution2 analysis per groups of index disease (cancer types of chronic diseases)
---

VERSION 1 – AUTHOR RESPONSE

RESPONSE TO COMMENTS FROM REVIEWER 1

Comment: To understand the background behind this paper that high income is related to high ICE, more information will be needed on how much is " a small fraction" is there any difference by income level?

Response: This is a very good point, for the reader to understand the findings we need to clarify the context, particularly, out-of-pocket costs of healthcare for individuals. In Sweden, patient fees account for a small fraction 3-5% of total health expenditure and there are nationally set cost ceilings of out-of-pocket patient fees, the max of 1,105 SEK on visits to healthcare services in a 12-month period, when this threshold is exceeded, there is no further patient fees for healthcare visits during the next 12-months, and there are separate cost ceilings set for prescription drugs. Additionally, the patient fees for a stay in inpatient care is 100 SEK per day per adult.

Action: the description of the Swedish Health and Social care system has been expanded and includes more detail on the out-of-pocket costs, this change can be found in the Introduction page 3.

Comment: Is there any information on the place of death and how long they stay at the hospital? They are important information.

Response: Yes, there is information on "Place of death" in the Cause of Death Register, the research team debated including this variable in the study given the limited precision of the variable in the cause of death register. As place of death is categorized as dying in hospital, in institutional care/ specialized geriatric clinic, private residence and other/unknown.

The length of hospital stay can be calculated using admission and discharge dates in the inpatient care register of The Region Stockholm Healthcare Administrative Database. However, for the purposes of this study the authors wanted to focus on inpatient care expenditures and as length of stay would be highly correlated with inpatient care expenditure

Action: The variable place of death is described in the Methods section on page 4 and is depicted in the results section in table 1 and the study population is stratified by place of death and the results are discussed in the Discussion section on page 14.

Comment: Is the hospital stay included in Months of institutional care?

Response: No, stays in hospital were not included in months of institutional care. The variable months in institutional care was measured in the Swedish Social Services Register and the use of social services (home-help and residency in institutional care) are reported on the last day of each month. Accordingly, we measured the months in institutional care in the last year of life based on an individual's date of death and the months they were registered as residences of an institution.

The Swedish Social Services Register only records the use of municipal social care, and the Region Stockholm Healthcare Administrative Database which records healthcare and hospital use. Although, there is onsite primary care provided in institutional care, unfortunately, we are not able to measure how many times as a resident has interacted with a doctor.

Comment: "Country of birth " should be like " birth outside or inside Sweden" because no info about other countries

Response: The country of birth variable is now referred to as “Born in Sweden” and “Born outside of Sweden”.

Action: This change can be throughout the text in the “Methods, Results, and Discussion” and in the tables.

Comment: What are others in social care?

Response: In this study, we measured the use of municipal social care as residing in institutional care (for the entire last year of life) or receiving home-help (domestic and/or personal care) in their own homes in the community. However, we found a group of individuals who were receiving home-help in their own homes in the community and who moved into institutional care during their last year of life, this group was referred to as the “Transition” group in the manuscript.

Action: The description of how municipal social care was measured has been explained in more detail in the Methods section page 6.

Comment: the result that advanced home healthcare users have lower ICE is very interesting. In the advanced home healthcare, more medical care may be offered and can be avoided hospitalization?

Response: Yes, the effect of home-healthcare is a very interesting finding. The research team has also considered what these findings indicate, such as could the provision of home-healthcare be effective in meeting the needs of extremely ill patients by providing necessary medical treatment in their homes and thus, reducing unplanned hospitalisations.

Comment: More explanation will be helpful including the payment of these two types of home care.

Response: Basic home-healthcare is free for all patients enrolled and for persons 85 years and older, so these patients are not required to pay fees for the care and treatment that is provided by the healthcare professionals responsible for performing basic home-healthcare in region Stockholm. Further, patients who receive advanced home-healthcare are also exempted from the patient fees for the treatment and care provided in their homes.

Action: The description of patient’s fees for basic and advanced home-healthcare patient fees is described in the Methods section on page 5.

Comment: " home-help users -- compared to those independent". This comparison is right? Home-help users should be always independent?

Response: Yes, you are correct that home-help users are independent as they live in their own homes in the community. In this study we used the term “independent” to indicate individuals who live in their own homes but do not receive home-help services.

We do consider home-help users and those independent to be comparable groups because both are living in their own homes in the community, with the distinction that home-help users have sought and are receiving formal support for care needs from municipality which enables them to remain living in their own homes. Although, the group described as independent did not receive municipal social care in their last year of life.

RESPONSE TO COMMENTS FROM REVIEWER 2

Comment: Analysis for place of death hospital versus home/home care institution

Response: Yes, there is information on “Place of death” in the Cause of Death Register, the research team debated whether to include this variable in the study, as place of death is categorized as dying in hospital, in institutional care/specialized geriatric clinic, private residence and other/unknown. However, as both reviewers have made enquiries about place of death, the variable has been included in the manuscript in a descriptive capacity.

Action: The variable place of death is described in the Methods section on page 4 and is depicted in the results section in table 1 and the study population is stratified by place of death and the results are discussed in the Discussion section on page 14.

Comment: Analysis per groups of index disease (cancer types of chronic diseases)

Response: Though an analysis focused on specific diseases would be interesting, there are limitations to completing such an analysis given our study population of persons 65 years and older in their last year of life, where the majority experienced multiple chronic conditions which makes it difficult to develop specific disease categories when an individual could be included in multiple disease categories.

As we understand that the underlying cause of death is a variable of interest in the manuscript, we did describe the underlying causes of death in table. However, we did not place a great emphasis on the underlying cause death because older people generally die with multiple chronic conditions and there are difficulties in discerning the exact cause of death especially for the very old. Additionally, autopsy is not frequently performed particularly older people [17]. Therefore, we considered the CCI score as a measure of morbidity and an indicator of need, to be more important variable to consider when investigating the outcome of expenditure which was incurred as a result of healthcare utilization.

VERSION 2 – REVIEW

REVIEWER	Nanako Tamiya University of Tsukuba, Health Services Research
REVIEW RETURNED	23-May-2022
GENERAL COMMENTS	They improved the manuscript following my comments well. Thank you for this opportunity.